# Implicit Extended Kalman Filter for Optical Terrain Relative Navigation Using Delayed Measurements

Stefano Silvestrini *, Margherita Piccinin, Giovanni Zanotti, Andrea Brandonisio, Paolo Lunghi and Michèle Lavagna

Department of Aerospace Science and Technologies, Politecnico di Milano, Via La Masa 34, 20156 Milan, Italy
* Correspondence: stefano.silvestrini@polimi.it

**Abstract:** The exploration of celestial bodies such as the Moon, Mars, or even smaller ones such as comets and asteroids, is the next frontier of space exploration. One of the most interesting and attractive purposes from the scientific point of view in this field, is the capability for a spacecraft to land on such bodies. Monocular cameras are widely adopted to perform this task due to their low cost and system complexity. Nevertheless, image-based algorithms for motion estimation range across different scales of complexities and computational loads. In this paper, a method to perform relative (or local) terrain navigation using frame-to-frame features correspondences and altimeter measurements is presented. The proposed image-based approach relies on the implementation of the implicit extended Kalman filter, which works using nonlinear dynamic models and corrections from measurements that are implicit functions of the state variables. In particular, here, the epipolar constraint, which is a geometric relationship between the feature point position vectors and the camera translation vector, is employed as the implicit measurement fused with altimeter updates. In realistic applications, the image processing routines require a certain amount of time to be executed. For this reason, the presented navigation system entails a fast cycle using altimeter measurements and a slow cycle with image-based updates. Moreover, the intrinsic delay of the feature matching execution is taken into account using a modified extrapolation method.

**Keywords:** optical navigation; epipolar constraints; Kalman filter; relative terrain navigation; features-based method; visual odometry; implicit constraints



## 1. Introduction and Background

Vision-based navigation is becoming the most prominent solution for autonomous navigation for different scenarios [1–5]. Moreover, activities linked to the Lunar Gateway and Lunar villages have created a deep interest in the lunar environment [6–8]. Numerous future missions will entail landing operations to deploy technological systems on the surface of the Moon. The driving precision landing requirement for the autonomous landing is to land within ~100 m of a predetermined location on the lunar surface, or any other planetary surface [9]. Traditional lunar landing approaches based on inertial sensing do not have the navigational precision to meet this requirement. The purpose of terrain navigation is to augment inertial navigation by providing position or bearing measurements relative to known surface landmarks, if available, or local motion estimate in unseen environments. From these measurements, the navigational precision can be reduced to a level that meets the ~100 requirement. According to [9], there are mainly two different navigation functions: global position estimation, here referred as absolute navigation and local position estimation, here referred as relative navigation. These functions can be achieved with active range sensing or passive imaging. Among all the different alternatives presented in the comprehensive review paper in [9], we can point out the following high-level clusters of approaches depending on the availability of a landmark/feature database:

- **Available database**: we refer to a database when a set of known landmarks or features of the viewed terrain is cataloged, reporting their absolute location with respect to a planetocentric reference frame, e.g., ME lunar frame. The easiest example is the crater matching approach, in which known lunar craters are cataloged with their latitude and longitude with respect to a Lunar fixed frame. The detection and matching of one of the cataloged landmarks (or computer-based database features) yield a bearing measurement that can be processed to establish the global position of the spacecraft. This approach is robust and accurate but it requires at least a partial knowledge of the terrain before the spacecraft sees it.

- **Unavailable database**: whenever the spacecraft is flying across unknown terrains, it is impossible to establish its global position from images only. Nevertheless, local (or relative) position, is still possible using frame-to-frame methods, which generally falls into the visual odometry domain in terrestrial robotics. Methods that fall within this classification differ between each other for increasing complexity level, along with increasing computational effort. In particular, the retrieval of a three dimensional description of the scene, e.g., the creation of a map in structure-from-motion-like methods, represent one of the major trade-off to be performed when assessing the feasibility of on-board implementation. These methods solve the motion estimation task by generating a 3D sparse map. A set of 2D-to-2D correspondences is obtained, relative pose between the frames is calculated and a sparse map of 3D points is initialized exploiting triangulation. The 2D features are then tracked for each subsequent frame and correlated to the 3D map: this way a set of 3D to 2D correspondences is obtained and used to solve the perspective-n-point problem, which along with a RANSAC routine set to delete incoming outliers (wrong match between target image and map), gives as a result a first estimate of the relative position of the camera.

In this paper, relative terrain navigation relies on frame-to-frame (2D-2D) motion estimation that works with extracted features from a pair of images, without creating any maps of the surrounding environment, coupled with an altimeter measurement. The reader is suggested to refer to [10] to deepen the knowledge on multi-view geometry and motion reconstruction. Once the features are tracked, a set of correspondences is available between the two frames. Feature correspondences are related by epipolar geometry [10], which completely describes the structure of the two consecutive camera poses and of the world points seen by them. This description is enclosed inside the essential matrix **E** for calibrated cameras, while fundamental matrix **F** holds for uncalibrated ones. Once one of these two matrices is available, motion can be retrieved up to scale by a simple algebraic decomposition. For particular cases in which the viewed scene is planar, feature correspondences may be also related by an homography matrix **H**, from which motion can again extrapolated up to scale. Among the frame-to-frame alternatives that can be employed for on-board applications in relatively low-power devices [11], two alternatives represent the state-of-the-art in the literature.

### 1.1. Normalized 8-Point Algorithm

The normalized 8-point algorithm developed by Richard Hartley [10] is one of the simplest and widely diffused method to obtain a fundamental matrix **F** from a set of 8 features correspondences without knowing the intrinsic parameters of the camera. The algorithm simply involves the construction and least square solution of a set of linear equations, and uses normalized input data for better conditioning of the problem and stability of the result. The suggested normalization is a translation and scaling of each image so that the centroid of the reference points is at the origin of the coordinates and the RMS distance of the points from the origin is equal to $\sqrt{2}$. It is known [10] that for the fundamental matrix **F** the following constraint is always valid: This gives rise to a set of linear equations of the form $\mathbf{A}f = 0$. If A has rank 8, then it is possible to solve for $f$ up to scale, and this is the classical way of implementing the 8-point algorithm. In the case where the matrix A has rank seven, it is still possible to solve for the fundamental

matrix by making use of the singularity constraint. The matrix **F** found by solving the set of linear equations will not in general have rank 2 (i.e., will not be exactly singular due to presence of noise in the extracted features $x_{k-1}, x_k$), consequently, steps to enforce this constraint are taken. The most convenient way to do this is to correct the matrix **F** found by the SVD solution from A. Matrix **F** is replaced by the matrix $F'$ that minimizes the Frobenius norm subject to the condition $det(F') = 0$. In the normalized 8-point algorithm singularity constraint is enforced solving directly for $F'$ using SVD, which is simple and rapid. Exact matrix **F** is then finally retrieved denormalizing matrix $F'$ obtained from the normalized data. This method is direct and efficient to compute, but has anyway a major drawback, which is associated to **F** matrix itself. Independently from the method used in fact, fundamental matrix suffers from the so called "planar structure degeneracy". If the observed points from the camera lies on a plane (as it could be the case during a landing phase, where surface is seen from far away and appears to be flat) **F** is determined only up to three degree of freedom, which leads to a three-parameter family of possible fundamental matrices F (one of the parameters accounts for scaling the matrix so there is only a two-parameter family of homogeneous matrices). This ambiguity is not solvable and is a consequence of the fact that the camera intrinsic (**K** matrix) are not included in **F**.

### 1.2. 5-Point Algorithm

Given a set of 2D feature correspondences, the most efficient solution to estimate the essential matrix **E** is represented by the five-point algorithm [10,12]. The problem is to find the possible solutions for relative camera pose between two calibrated views given five corresponding points. The algorithm consists of computing the coefficients of a tenth degree polynomial in closed form and subsequently finding its roots. Only relative positions of the points and cameras can be recovered, overall scale of the configuration represents an ambiguity and can never be extrapolated solely from images. Image points are represented by homogeneous 3-vectors in the first and second view, respectively. World points are represented as homogeneous 4-vectors X. This algorithm is thought to be used in conjunction with pre-emptive RANSAC in order to be more robust to the presence of outliers (i.e., false matches, wrong tracking) between the features. A number of random samples are taken, each containing five points. Five-point algorithm is applied to each sample and thus a number of hypotheses are generated. The best hypothesis is then chosen according to a robust measure over all the tracks and is in the end iteratively refined. One of the most important features of the five-point algorithm is that it works for any kind of configuration of the features considered, avoiding the planar structure degeneracy. This algorithm results to be efficient both in terms of accuracy and speed. In comparison to other state-of-the-art methods, despite being weaker for determining rotations, five-point works optimally for sideways motion and similarly for forward motion, although slightly worse when baseline between the two cameras is very small. Using the essential matrix also removes the projective ambiguity, which arises using fundamental matrix F, and provides a metric (or singular) reconstruction, which means the 3D points are true up to scaling alone, and not up to a projective transformation. Ideally, the output of the correspondent locations $\vec{m}_{k-1}, \vec{m}_k$ and the essential matrix is enough for a navigation filter to perform state estimation. As the number of feature grows, it may be convenient to pre-process such information to recover relative rotation and translation directly in the image processing block.

In principle the above algorithms delivers a relative pose of consecutive views of the camera, in the form of essential matrix for instance. In this way, direct relative translational vector would be input to the filter.

In this paper, the implicit extended Kalman filter (IEKF) receives a set of normalized Euclidean correspondences and the essential matrix. This vision-based approach is realized via an implementation of the IEKF, which is a variation of the classical Kalman filter that allows incorporating implicit functions as measurement constraints of the state variables. The epipolar constraint, which is a geometric relationship between the feature point position

vectors and the camera translation vector, is employed as the implicit measurement in the Kalman filter. Basically, the algorithm is based on the epipolar constraint, which is manipulated by matrix decomposition to obtain relative translation and rotation. The core of this paper is to develop an algorithm that can be deployed on-board. The problem of deterministic delay due to the image processing time is taken into account using a customized extrapolation method that can fuse the high-frequency altimeter measurements with the low-frequency optical information, with an intrinsic delay of 1 s (Figure 1).

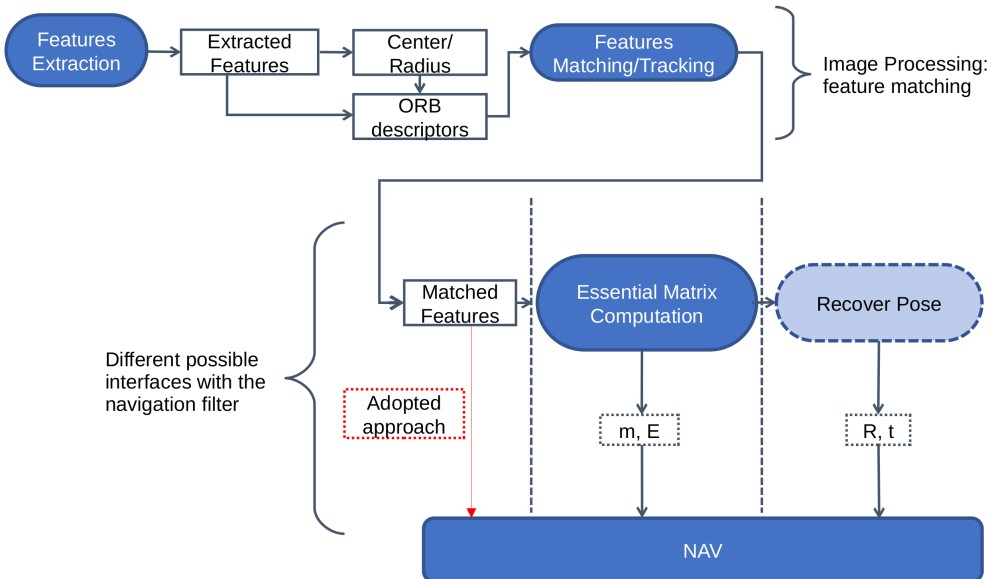

**Figure 1.** Schematics of the required pre-processing steps to interface with the navigation filter. The figure shows the different possibilities for interfacing the feature-based image processing with the navigation filter with unknown landmarks and no 3D-map generation. In particular, the main difference is whether the essential matrix is reconstructed externally (e.g., 5-point algorithm) and used to retrieve the relative pose or it is embedded in the filter as the implicit measurement function.

## 2. Navigation Algorithm Description

The relative navigation reconstructs the relative state with respect to the defined ground reference system (GRS). The idea is to detect salient features, which may not be part of a database, and track them in subsequent frames as outlined in Figure 2.

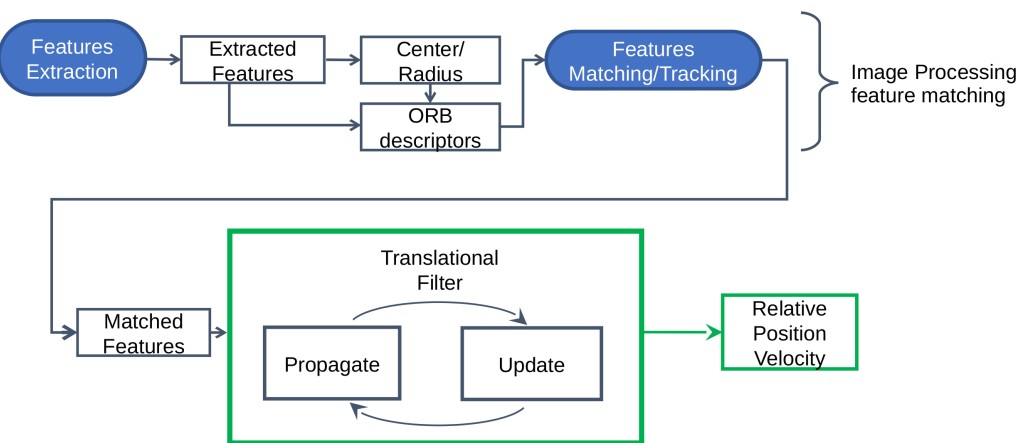

**Figure 2.** Relative terrain navigation algorithm. The figure relates to Figure 1 showing the adopted baseline for this work in terms of IP-navigation interfaces.

## 3. Feature Detection & Matching

The pinhole camera model is adopted in this paper. Given the location of a detected feature $\rho_{f,\mathbf{B}}$ in the camera frame, which is equal to the body frame $\mathbf{B}_{b_1,b_2,b_3}|_k$ for simplicity:

$$\vec{\rho_f} = \begin{bmatrix} x \\ y \\ z \end{bmatrix}_{\mathbf{B}} \in \mathcal{R}^3 \tag{1}$$

the pinhole camera model expresses the location of the feature point in the focal plane as:

$$\vec{m} = \begin{bmatrix} u \\ v \\ 1 \end{bmatrix} \tag{2}$$

where:

$$u = \frac{x}{z}, v = \frac{y}{z} \tag{3}$$

The ORB-descriptors are employed to make the matching more robust. ORB descriptors assign an orientation to each feature such as left or right facing depending on how the levels of intensity change around that feature. For detecting intensity change, ORB uses intensity centroid. Given the assumption that a corner's intensity is offset from its center, the intensity centroid may be used to impute an orientation. To prevent the descriptor from being sensitive to high-frequency noise, BRIEF method smooths image using a Gaussian kernel. Subsequently, a pair of pixels is selected in a defined neighborhood around that feature. The defined neighborhood around pixel is known as a patch, which is a square of some pixel width and height.

## 4. Relative Navigation Filter

Algorithm 1 reports the most relevant steps of the EKF procedure in a pseudo-code format. The extended Kalman filter in Algorithm 1 represents the basis for the implementation of the algorithm; nevertheless, strong modifications have been made in the update step and the asynchronous measurement integration, described in Sections 4.2 and 4.4.

---

**Algorithm 1** Extended Kalman Filter

1: $\hat{\vec{x}}_k^- = \int_{t_{k-1}}^{t_k} f(\vec{x}(\tau))d\tau,$      $\vec{x}_{k-1} = \hat{\vec{x}}_{k-1},$      $\hat{\vec{x}}_0^+ = \vec{x}_0$     ▷ Prediction step

2: $\mathbf{F}_k = \frac{\partial f}{\partial \vec{x}}\Big|_{\hat{x}_{k-1}},$      $\mathbf{H}_k = \frac{\partial \mathbf{h}}{\partial \vec{x}}\Big|_{\hat{x}_k}$     ▷ State and measurement Jacobian matrices

3: $\mathbf{\Phi}(t_k, t_{k-1}) = \mathbf{I}_{6\times6} + \mathbf{F}_k \Delta t$     ▷ State Transition Matrix

4: $\mathbf{P}_k^- = \mathbf{\Phi}(t_k, t_{k-1})\mathbf{P}_{k-1}^+\mathbf{\Phi}^T(t_k, t_{k-1}) + \mathbf{Q},$   $\mathbf{P}_0^+ = \mathbf{P}_0$     ▷ State Covariance matrix propagation

5: $\mathbf{K}_k = \mathbf{P}_k^-\mathbf{H}_k^T(\mathbf{H}_k\mathbf{P}_k^-\mathbf{H}_k^T + \mathbf{R}_k)^{-1}$     ▷ Kalman gain matrix computation

6: $\hat{\vec{x}}_k^+ = \hat{\vec{x}}_k^- + \mathbf{K}_k(\vec{y}_k - h(\hat{\vec{x}}_k^-))$     ▷ Correction step

7: $\mathbf{P}_k^+ = (\mathbf{I} - \mathbf{K}_k\mathbf{H}_k)\mathbf{P}_k^-(\mathbf{I} - \mathbf{K}_k\mathbf{H}_k)^T + \mathbf{K}_k\mathbf{R}\mathbf{K}_k^T$     ▷ State Covariance matrix correction

---

### 4.1. Prediction Step

The prediction step of the implicit Kalman filter follows the classical formulation of Algorithm 1. It is important to remark that the distinction between absolute and relative navigation is the reference frame in which the state is reconstructed. In the absolute navigation, the state of the spacecraft is reconstructed with respect to the lunar fixed frame. The lunar fixed frame considered is the mean Earth/polar axis (ME) reference system, as shown in Figure 3, described in details in [13,14].

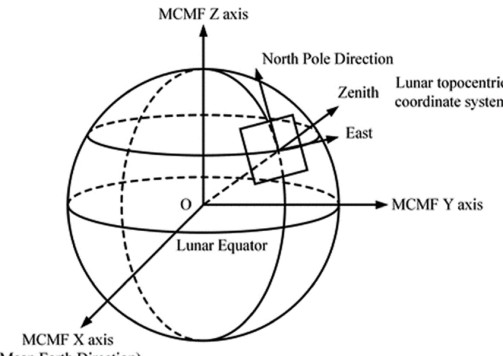

**Figure 3.** Lunar fixed frame mean Earth/polar axis. Figure Credits [13].

It defines the z-axis as the mean rotational pole. The prime meridian (0 degrees Longitude) is defined by the mean Earth direction. The intersection of the lunar equator and prime meridian occurs at what can be called the Moon's "mean sub-Earth point". The concept of a lunar "sub-Earth point" derives from the fact that the Moon's rotation is tidally locked to the Earth. The actual sub-Earth point on the Moon varies slightly due to orbital eccentricity, inclination, and other factors. So, a "mean sub-Earth point" is used to define the point on the lunar surface where Longitude equals 0 degrees. This point does not coincide with any prominent crater or other lunar surface feature. During these phases, orbital mechanics equations govern the motion. In case of relative navigation, the ground reference system (GRS) is used, depicted in Figure 4.

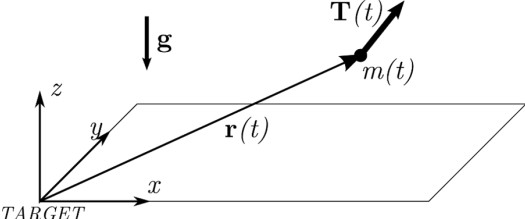

**Figure 4.** Ground Reference System [15].

In the case of a close approach during landing, distances, for both downrange and altitude, are small compared to the planet's radius; thus, the assumption of a constant gravity field with flat ground is appropriate. This assumption is widely used and accepted in the literature (see [16]). The translational dynamics of the spacecraft are expressed in a ground reference system (GRS), where z-axis is the altitude, x-axis downrange in the direction of flight and the consequent y-axis cross-range. The dynamics during the power descent phase is described by the equations:

$$
\begin{cases}
\dot{\vec{r}} = \vec{v} \\
\dot{\vec{v}} = \dfrac{\vec{T}}{m} + \vec{g} \\
\dot{m} = -\dfrac{T}{I_{sp} g_0}
\end{cases}
\tag{4}
$$

where $T$ is the thrust vector delivered by the on-board propulsion, whose specific impulse is $I_{sp}$. In synthesis, the filter state can be written as:

$$
\vec{x} = \begin{bmatrix} \vec{r} \\ \vec{v} \end{bmatrix}
\tag{5}
$$

The dynamical model is linear, hence the linearization is not required to compute the covariance propagation as:

$$
\mathbf{P}_k^- = \mathbf{\Phi}(t_k, t_{k-1}) \mathbf{P}_{k-1}^+ \mathbf{\Phi}^\mathsf{T}(t_k, t_{k-1}) + \mathbf{Q}, \quad \mathbf{P}_0^+ = \mathbf{P}_0
\tag{6}
$$

where the state transition matrix can easily be written as:

$$\mathbf{\Phi}(t_k, t_{k-1}) = \mathbf{I}_{6\times6} + \begin{bmatrix} \mathbf{0}_{3\times3} & \mathbf{I}_{3\times3} \\ \mathbf{0}_{3\times3} & \mathbf{0}_{3\times3} \end{bmatrix} \cdot (t_k - t_{k-1}) \tag{7}$$

This paper does not investigate thoroughly the influence of the dynamical model on the performance of the relative terrain estimation. The novelty lies on the processing of the frame-to-frame matched features. Regarding the dynamics to be employed, more accurate environmental representation can be used, such as the one described in reference [1]. A more accurate dynamical model would certainly be beneficial for the filter accuracy, nevertheless literature mostly reports applications in which constant gravity field is used for planetary landing [17]. The introduction of the intrinsic formulation coming from the epipolar constraint allows the filter to be directly fed with the frame-to-frame matching without the need for an additional pre-processing step to retrieve relative translation from sequential images (e.g., using the 5-points algorithm and essential matrix decomposition). The increase in time for propagating the a priori state with a more complex dynamics is negligible with respect to the above-mentioned pre-processing step.

*4.2. Implicit Measurement Function: The Epipolar Constraint*

The core of this paper is the development of an estimation algorithm that incorporates matched features directly, without augmenting the internal states of the filter. The state of the filter $\vec{x}$ is the estimate of the relative position $\vec{r}$ and velocity $\vec{v}$ vectors. The filter receives two elements for each matched features as measurement. Indeed, the locations of the $n_f$ matched features in the two successive camera frames ($k-1$ and $k$) are stored in stacked variables, $\vec{m}_{k-1}$ and $\vec{m}_k$, whose columns:

- $\vec{m}_{k-1,i} = [u_{k-1} \ v_{k-1} \ 1]_i^{\mathsf{T}}$: location of the matched feature $i$ in camera frame at time instant $k-1$;
- $\vec{m}_k, i = [u_k \ v_k \ 1]_i^{\mathsf{T}}$: location of the matched feature $i$ in camera frame at time instant $k$.

The frame-to-frame matched features represent the measurements at each time step. Nevertheless, it is not trivial to obtain a measurement function that links the filter states $\vec{r}$ and $\vec{v}$ to the matched features between two consecutive frames $k-1$ and $k$. In principle, one would need an explicit function in the form of $\vec{m}_{k-1}(\vec{x}_{k-1})$, $\vec{m}_k(\vec{x}_k)$. Each of these features are related by the epipolar constraint given by Equation (8).

$$0 = \vec{m}_{k,i}^{\mathsf{T}} \mathbf{E} \vec{m}_{k-1,i} \tag{8}$$

where

$$\mathbf{E} = \mathbf{R}[\vec{t}]_\times \tag{9}$$

in which $\mathbf{R}$ is the rotation matrix between the camera poses in two successive frames, namely from step $k-1$ to $k$ and $\vec{t}$ is the translation between the camera origin between two successive frames, expressed in the camera frame at step $k$, which is equal to the body frame $\mathbf{B}_{b_1,b_2,b_3}|_k$ for simplicity. Typical space applications require the attitude to be reconstructed with respect to one inertial frame or, at least, a planetocentric reference frame. As mentioned, this paper investigates only relative position and velocity estimation from optical measurement; thus, it is assumed that an internal knowledge of the system attitude with respect to an inertial frame $\mathbf{I}_{i,j,k}$ is known. The reference frames, relevant for visual odometry, are described in Figure 5. With reference to Figure 5, one can write the epipolar constraint as:

$$\mathbf{R} = [\mathbf{B}\mathbf{R}_{\mathbf{I},k}][\mathbf{B}\mathbf{R}_{\mathbf{I},k-1}]^{\mathsf{T}} \tag{10}$$

$$\vec{t} = (\vec{r}_k - \vec{r}_{k-1})|_k \tag{11}$$

where it is important to note that the translation between two camera frames is expressed in the body frame $\mathbf{B}_{b_1,b_2,b_3}|_k$, at time instant $k$.

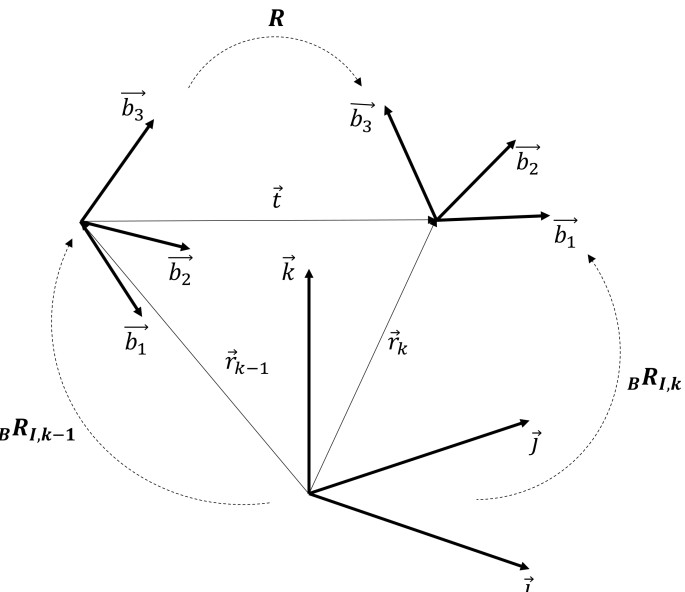

**Figure 5.** Frame-to-frame visual odometry reference frames.

With the above definitions, following the development of [18,19], we can rearrange the epipolar constraint in Equation (8), recalling that we have $n_f$ epipolar constraints, each one following Equation (8). Thus, rearranging:

$$0 = \mathbf{C}_{k-1,k}\vec{e} \tag{12}$$

where $\mathbf{C}_{k-1,k} \in \mathcal{R}^{n_f \times 9}$ is a matrix, whose rows correspond to the following vector referred to a given matched feature:

$$\mathbf{C}_{k-1,k} = \begin{bmatrix} \vec{c_1} \\ \vdots \\ \vec{c_{n_f}} \end{bmatrix} \rightarrow \vec{c_i} = [u_k u_{k-1} \ v_k u_{k-1} \ u_{k-1} \ u_k v_{k-1} \ v_k v_{k-1} \ v_{k-1} \ u_k \ v_k \ 1] \tag{13}$$

and $\vec{e} = [e_{11} \ e_{21} \ e_{31} \ e_{12} \ e_{22} \ e_{32} \ e_{13} \ e_{23} \ e_{33}]^\mathsf{T}$ is a vector of stacked columns of the epipolar matrix $\mathbf{E}$. From the definition of the feature matrix $C$ and the epipolar matrix $E$, it is straightforward to note the variable dependencies of such matrices. In other words, one can write:

$$\mathbf{C}_{k-1,k} = \mathbf{C}(\vec{m}_{k-1}, \vec{m}_k) \tag{14}$$
$$\mathbf{E} = \mathbf{E}(\vec{r}, \vec{q}) \tag{15}$$

where $\vec{q}$ is the quaternion state, which is typically the attitude representation available on-board. By recalling that the filter state $\hat{\vec{x}} = [\hat{\vec{r}}, \hat{\vec{v}}]$ we can explicitly write the implicit measurement as a function of the filter state. Using the above derivation, we can finally express the measurement that will be processed in the implicit Kalman filter:

$$\hat{\vec{y}}_k = h(\hat{\vec{x}}_k, \vec{m}_{k-1}, \vec{m}_k) = \mathbf{C}_{k-1,k}(\vec{m}_{k-1}, \vec{m}_k) \cdot \vec{e}(\hat{\vec{r}}_k, \vec{q}_k) \tag{16}$$

It is important to note that the true value of Equation (16) is always $\vec{y}_k = 0$ because it is a geometric constraint that is always satisfied. Using the estimated state to compute the essential matrix yields an innovation term that is defined as the difference between the true value $\vec{y}_k = \vec{0}$ and the implicit measurement $\hat{\vec{y}}_k$.

### 4.3. Correction Step

The implicit measurement derived in Section 4.2 is used in the correction step of the relative terrain navigation filter. In particular, with reference to Algorithm 1, the correction steps entails the implicit measurement $\hat{\vec{y}}_k$ calculated using the current state estimate $\hat{\vec{x}}_k$ and the set of matched features $\vec{m}_{k-1}$ and $\vec{m}_k$:

$$\hat{\vec{y}}_k = h(\hat{\vec{x}}_k, \vec{m}_{k-1}, \vec{m}_k) = \mathbf{C}_{k-1,k}(\vec{m}_{k-1}, \vec{m}_k) \cdot \vec{e}(\hat{\vec{x}}_k, \vec{q}_k) \tag{17}$$

The measurement update in the correction step of Algorithm 1 can be written as:

$$\hat{\vec{x}}_k^+ = \hat{\vec{x}}_k^- + \mathbf{K}_k(\vec{y}_k - h(\hat{\vec{x}}_k^-, \vec{m}_{k-1}, \vec{m}_k)) \tag{18}$$

$$= \hat{\vec{x}}_k^- - \mathbf{K}_k h(\hat{\vec{x}}_k^-, \vec{m}_{k-1}, \vec{m}_k) \tag{19}$$

$$= \hat{\vec{x}}_k^- - \mathbf{K}_k(\mathbf{C}_{k-1,k}(\vec{m}_{k-1}, \vec{m}_k) \cdot \vec{e}(\hat{\vec{x}}_k, \vec{q}_k)) \tag{20}$$

The Kalman gain in Equation (20) is calculated through the standard formula using the measurement covariance matrix first order approximation $\tilde{\mathbf{R}}_k$. This is needed due to the implicit form of the measurement function, which derives from the noise covariance matrix of the measured feature localization $\mathbf{R}_k$. In particular the Kalman gain can be written:

$$\mathbf{K}_k = \mathbf{P}_k^- \mathbf{H}_k^T (\mathbf{H}_k \mathbf{P}_k^- \mathbf{H}_k^T + \tilde{\mathbf{R}}_k)^{-1} \tag{21}$$

where the first-order approximation of the measurement covariance matrix is derived:

$$\tilde{\mathbf{R}}_k = \mathbf{D}_k \mathbf{R}_k \mathbf{D}_k^\mathsf{T} \tag{22}$$

in which the diagonal matrix:

$$\mathbf{D}_k = \frac{\partial \vec{y}_k}{\partial \tilde{\vec{Y}}_k} = \frac{\partial \mathbf{C}_{k-1,k}(\vec{m}_{k-1}, \vec{m}_k)}{\partial \tilde{\vec{Y}}_k} \cdot \vec{e}(\hat{\vec{x}}_k, \vec{q}_k) \tag{23}$$

where the vector $\tilde{\vec{Y}}_k = [u_{1,k-1}\ v_{1,k-1}\ u_{1,k}\ v_{1,k}\ \ldots u_{n_f,k-1}\ v_{n_f,k-1}\ u_{n_f,k}\ v_{n_f,k}]^\mathsf{T}$ combines the locations of all the $n_f$ matched features at two consecutive steps. In order to complete the calculation of the Kalman gain in Equation (21), the Jacobian matrix of the measurement function $h$ needs to be computed as:

$$\mathbf{H}_k = \frac{\partial h_k}{\partial \vec{x}_k}\Big|_{\hat{\vec{x}}_k^-} = \mathbf{C}_{k-1,k}(\vec{m}_{k-1}, \vec{m}_k) \frac{\partial \vec{e}(\hat{\vec{x}}_k, \vec{q}_k)}{\partial \vec{x}_k}\Big|_{\hat{\vec{x}}_k^-} \tag{24}$$

The core correction step is represented by the optical measurements described so far. Nevertheless, landing navigation often uses high frequency altimeter measurements fused in the navigation filter. As mentioned, the altimeter frequency is eight times the optical one, namely $f_{alt} = 8$ Hz. This means that the filter updates every $t_{alt} = 125$ ms with altimeter measurements only and every $t_{opt} = 1$ s with a major correction steps including the optical images. For the selected filter state, the measurement matrix for the altimeter measurements is fairly simple, because the vertical distance with respect to the terrain is equal to the $z$ component of the position:

$$\vec{h}_{alt} = [0\ 0\ 1\ 0\ 0\ 0] \tag{25}$$

that represents the measurement vector to be used in the Kalman update. Such vector is used alone during the fast cycles, and appended to the matrix $\mathbf{H}_k$ of Equation (24) when the major slow cycle occurs. The issue with asynchronous updates due to the image processing computational time is explored in the next Section 4.4.

### 4.4. Delayed Measurements Integration

The navigation algorithm heavily relies on optical measurements. The information content is extracted from the images through the feature extraction and matching algorithms and the intermediate post-processing. Such process takes a finite amount of time that needs to be taken into account when fusing the measurements in the IEKF, especially for real-time applications. Indeed, when delayed measurements are present, at instant $k$ the system receives a delayed measurement corresponding to time instant $s$ ($s = k - N$, where $N$ number of delay samples). There are various methods to consider the measurements delays in the navigation filter [1]:

- Filter recalculation method: it consists of coupling two filters running at fast and slow rate [20]. The former incorporates the high-frequency measurements, whereas the latter is activated every time a delayed (e.g., slow and less frequent) measurement arrives. The method computes the entire trajectory of the state until the current step. Using this method, optimality is guaranteed at the cost of computational burden.
- Alexander method: it consists of updating the covariance matrices at time $s$ as if the delayed measurement arrived. Then, once measurements $\mathbf{Y}_s$ are inserted at time $k$, the update is simply the standard Kalman filter one with a correction matrix term [21].
- Larsen extrapolation method: The method described in [21] requires the measurement matrix $\mathbf{H}_s$ and the noise distribution matrix $\mathbf{R}_s$ at time $s$. In the presented scenario, this is not valid: the measurement matrix depends on the relative positioning of the camera and craters. Larsen developed a measurement extrapolation method that does not require knowledge about the two matrices until time $k$ [22]. This method is taken as a reference to implement a modified version suitable for the analyzed scenario.

The adaptation of the Larsen method for the measurement fusion is hereby described. For details on the derivation, the reader is suggested to refer to the original reference [22]. Several modifications were needed to solve two shortcomings of the original method: the incorporation of high-frequency altimeter and the extension to the nonlinear extended Kalman filter. For the former, the filter firstly computes the gain and the updates as in Algorithm 1 fusing fast altimeter measurements. For what concerns the delayed measurements, let us call the measurements coming from the time instant $s = k - N$ as $\vec{y}_s$, which are incorporated at time instant $k$. The Larsen method consists of calculating an extrapolated measurements from $\vec{y}_s$ to be integrated at time $k$, called $\vec{y}_{k,s}^{ext}$:

$$\vec{y}_{k,s}^{ext} = \vec{y}_s + h(\hat{\vec{x}}_k^-) - h(\hat{\vec{x}}_s^+) \tag{26}$$

where $\vec{y}_s = \vec{0}$ because it represents a geometric epipolar constraint at any time. At each intermediate step between $s$ and $k$ a correction term $M$ is calculated as:

$$\mathbf{M}_k = \left[ \prod_{i=0}^{k-s-1} (\mathbf{I} - \mathbf{K}_{k-i}\mathbf{H}_{k-i})\mathbf{\Phi}(t_{k-i}, t_{k-i-1}) \right] \mathbf{P}_s \tag{27}$$

where the Kalman gain and measurement sensitivity matrix $\mathbf{H_{k-i}}$ at step $k - i$ does not reflect any update coming from the delayed measurement $\mathbf{Y}_s$. Then, the updates of the correction term are calculated as follows, modifying the correction equations in Algorithm 1:

$$\mathbf{K}_{k,s} = \mathbf{M}_k \mathbf{H}_{k,s}^T [\mathbf{H}_{k,s}\mathbf{P}_s\mathbf{H}_{k,s}^T + \mathbf{R}_s]^{-1} \tag{28}$$

$$\hat{\vec{x}}_k^+ = \hat{\vec{x}}_k^- + \mathbf{K}_{k,s}(\vec{y}_{k,s}^{ext} - h(\hat{\vec{x}}_k^-)) \tag{29}$$

$$\mathbf{P}_k^+ = (\mathbf{I} - \mathbf{K}_{k,s}\mathbf{H}_{k,s}\mathbf{M}_k^T\mathbf{P}_k^{-1})\mathbf{P}_k^-(\mathbf{I} - \mathbf{K}_{k,s}\mathbf{H}_{k,s}\mathbf{M}_k^T\mathbf{P}_k^{-1})^T + \mathbf{K}_{k,s}\mathbf{R}_s\mathbf{K}_{k,s}^T \tag{30}$$

The covariance update is a modified version of the Joseph formula adapted to the original Larsen method. This is performed to ensure that the covariance matrix remains positive semi-definite. As seen in Equations (26) and (27), the extrapolation method always

requires only two matrix multiplications at each time instant and the storage of two variables any time an image is acquired.

## 5. Numerical Results

The following section provides preliminary insights on the algorithm performance during a sample landing trajectory.

### 5.1. Scenario Description

The considered mission scenario consists of the spacecraft descent on the lunar South Pole from an altitude of 100 km down to 3 km. The landing area, within the South Pole, is defined during the mission. A 2D planar Moon landing is taken as reference, nevertheless the approach is easily applicable to a 3D scenario (including cross-range direction). Spacecraft trajectories are generated executing optimal guidance algorithm depending on the target location and thrust constraints. Moreover, due to the given landing location, it is critical to take into account illumination and shadowing condition. In particular, previous studies have highlighted that for the landing site selection it is fundamental to consider the South Pole regions that present areas in sunlight. Since the angle between the Moon rotation axis and the ecliptic is about 90 degrees, in the Polar Regions the topography plays a crucial role for the illumination conditions. In fact, areas at relatively high altitude can experience continuous periods of illumination (of several months), whereas some crater bottoms are always in shadow. The Moon landing mission scenario shall aim at landing sites (LS) with such characteristics. In such scenario, the navigation system can encounter highly varying illumination conditions, with low Sun elevation angle in the South Pole region and large shadow areas in the image. Figure 6 shows the assumed nominal phases for a Lunar landing mission.

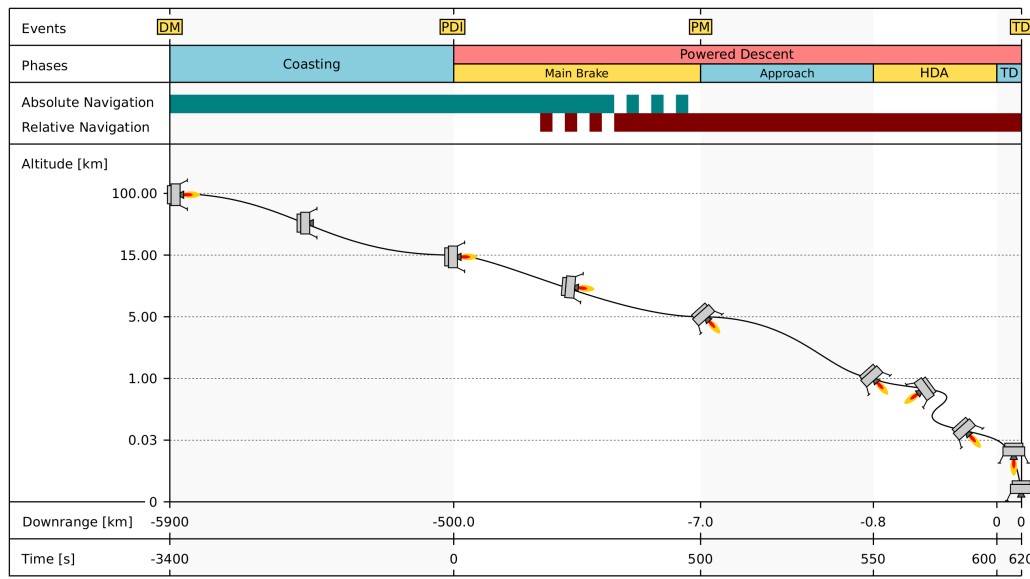

**Figure 6.** Nominal lunar landing phases and navigation modes. Distances and times are not in scale. The time scale takes the PDI as origin, while downrange is assumed to be 0 at the landing site [1].

It can be seen how on-board navigation operates in two modes, absolute navigation mode [1,14] and relative navigation mode, discussed in this paper. In the following, the landing phases are detailed:

- Parking orbit (PO): the spacecraft is in its orbital motion around the Moon and performs absolute navigation with respect to the lunar fixed frame. The parking orbit is assumed to be a circular orbit with constant altitude between 250 and 100 km. The altitude 100 km is the value assumed in the paper for the lunar pinpoint landing

scenario. Absolute navigation is performed. In traditional algorithms, lunar maps and craters catalog are used to determine position and velocity.

- Maneuver (DM): the spacecraft performs a tangential burn to lower the orbit perigee, inserting itself into an elliptical orbit. The lower the perigee, the lower the overall amount of fuel required for the landing maneuver. At the same time, the terrain topography pose a safety requirement on the minimum altitude of the perigee. Moreover, 15 km is a generally accepted value and is adopted as nominal value in this study.
- Coasting phase: the spacecraft follows the $100 \times 15$ km elliptical transfer orbit in ballistic flight. This phase ends after half orbit at the perigee. The spacecraft performs absolute navigation.
- Powered descent initiation (PDI): the coasting phase terminates at the transfer orbit perigee, nominally at 15 km altitude. At this point, the main thrusters are turned on and the powered descent starts. In the following, the main states of the spacecraft at PDI are listed:
    - Downrange: [–550, –450] km
    - Altitude: 15 km
    - Velocity: ~1700 m/s
    - Time to touchdown: [–600, –500] s
- Main brake phase: in this phase, the spacecraft drops most of its horizontal velocity. The thrust magnitude is constant and close to the maximum. The thrust vector pointing profile is optimized and remains close to the local horizon. During most of this phase the navigation is absolute, while in the last part relative navigation is initialized, for it is required to be running as the landing site comes into the camera field of view. At the end of the maneuver, the spacecraft is in the following state:
    - Downrange: [–15,000, –1500] m
    - Altitude: [7000, 2000] m
    - Velocity: [100, 60] m/s
    - Time to touchdown: [–100, –70] s
- Final approach phase: at the end of the main brake, the nominal landing site comes into the field of view of the navigation system. The constant thrust constraint is released and the spacecraft performs a pitch maneuver (PM) to point the thrust vector mainly toward ground. In this phase relative navigation is performed; the landing area is scanned and large diversions to the landing trajectory can be commanded to cope with errors. The state of the spacecraft at the end of the final approach is in the following ranges:
    - Downrange: [–800, –450] m
    - Altitude: [1500, 500] m
    - Velocity: [50, 20] m/s
    - Time to touchdown: [–40, –20] s
- Fine trajectory correction and hazard avoidance: below 1500 m of altitude, fine trajectory corrections (in the maximum order of magnitude of hundreds of meters) can be ordered to perform the hazard avoidance task. This phase ends on the vertical of the selected landing site at a certain altitude (tens of meters), with null horizontal velocity to ensure a vertical touch down.
- Terminal descent: powered vertical descent at constant velocity till the touchdown.

In an inertial reference frame, at the touchdown of the lander, horizontal velocity is required to match the velocity of the terrain due to the rotation of the Moon around its axis. Assuming a landing at the South Pole, the inertial tangential velocity of the lunar surface at the landing site can be neglected, in first approximation, and the overall landing trajectory is assumed to be planar (in the inertial reference frame). There are several strategies in the selection of sensors suite for relative and absolute navigation. The landing is performed using a camera with 40° field of view, a $1024 \times 1024$ pixel sensor, and an

altimeter. As summary, the nominal trajectory used for testing the navigation system is shown in Figures 7 and 8.

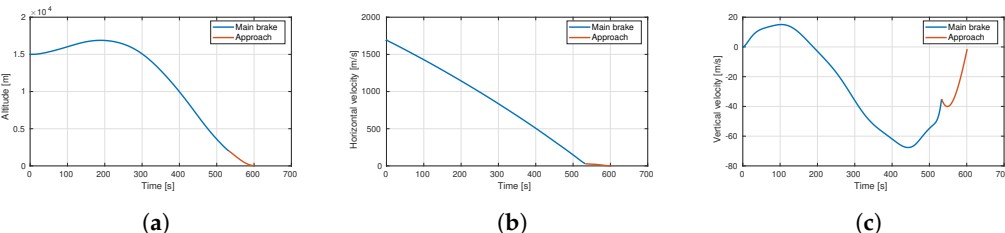

(**a**)          (**b**)          (**c**)

**Figure 7.** Altitude and velocity profile (horizontal and vertical) for the nominal landing trajectory (powered descent and final approach only). Same testing trajectory as [1]. (**a**) Altitude. (**b**) Horizontal velocity. (**c**) Vertical velocity.

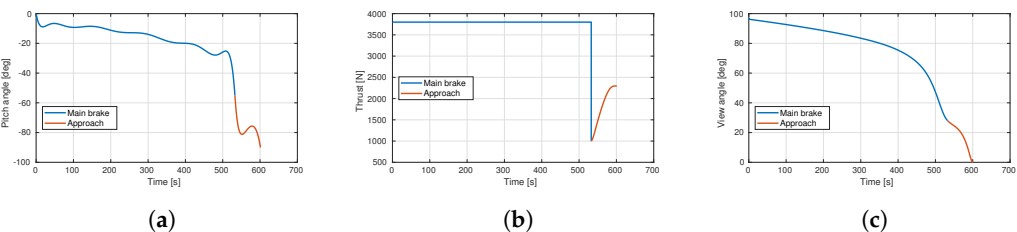

(**a**)          (**b**)          (**c**)

**Figure 8.** Thrust profile (orientation and magnitude) and pitch angle profile for the nominal landing trajectory (powered descent and final approach only). Same testing trajectory as [1]. (**a**) Thrust angle. (**b**) Thrust magnitude. (**c**) View angle over landing site.

The generation of synthetic images was performed using a custom rendering pipeline based on PANGU software, as shown in Figure 9 [23,24].

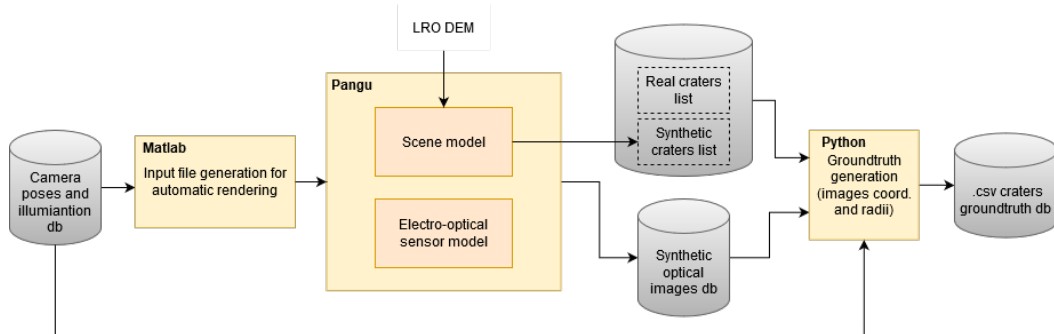

**Figure 9.** Synthetic images generation tool for relative terrain navigation testing.

## 5.2. Feature Detection and Matching

This paper does not focus on the features detection and matching pipeline, as a robust and common method is used to retrieve these elements from the images. The ORB detector implemented in OpenCV [25] is used to process images, while a brute-force matching approach based on Hamming distance is used to perform the frame-to-frame features matching. A sample frame-to-frame matching is shown in Figure 10. The feature extraction and description procedure implements the following:

```python
## Feature extraction
#-------------------
kpkm1 = orb.detect(imgOrbkm1, None)
kpk = orb.detect(imgOrbk, None)

# compute descriptors
kpkm1, deskm1 = orb.compute(imgOrbkm1, kpkm1)
kpk, desk = orb.compute(imgOrbk, kpk)
```

Once features are extracted from two consecutive images, the brute-force matcher object yields a set of corresponding features between the two consecutive frames. The set of matched features is ordered based on their Hamming distances, meaning that the most confident matches are ranked higher than less reliable ones. In coding, the instructions are as follows:

```python
## Feature matching
#------------------
# create BFMatcher object
bf = cv.BFMatcher(cv.NORM_HAMMING, crossCheck=True)

# match descriptors.
matches = bf.match(deskm1, desk)

# sort them in the order of their distance.
matches = sorted(matches, key=lambda x: x.distance)

## extract the matched keypoints
src_pts = np.float32([kpkm1[m.queryIdx].pt for m in matches]).reshape
                                            (-1, 1, 2)
dst_pts = np.float32([kpk[m.trainIdx].pt for m in matches]).reshape(-
                                            1, 1, 2)

if src_pts.shape[0] > N_FEAT:
    # store matched craters
    mkm1[frame_index-1, :N_FEAT, :] = np.squeeze(src_pts[:N_FEAT])
    mk[frame_index-1, :N_FEAT, :] = np.squeeze(dst_pts[:N_FEAT])
```

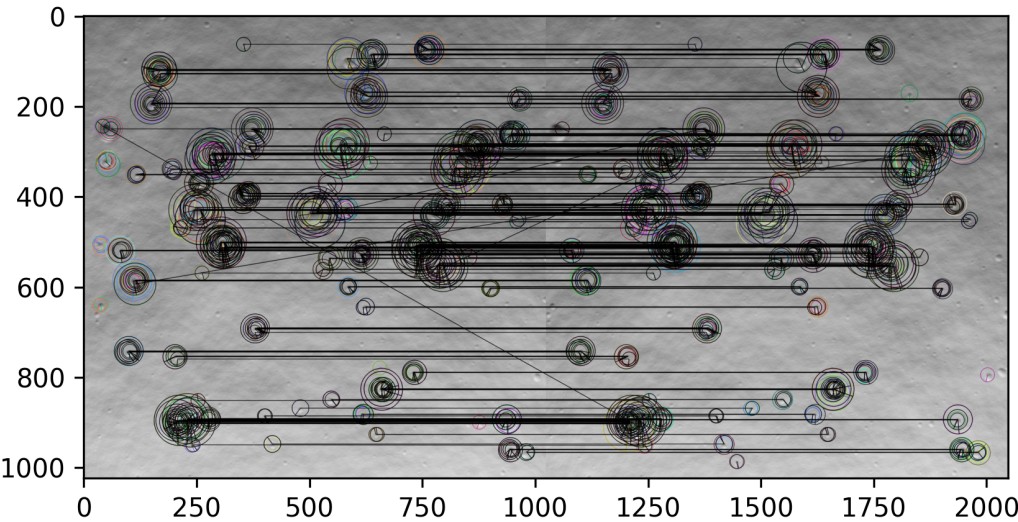

**Figure 10.** Frame-to-frame features detection and matching using ORB features.

The average number of tracked features in consecutive images is roughly 500. Nevertheless, the number of features influences the computational complexity of the filter. In particular, the size of the epipolar constraint in Equation (16) scales linearly with the number of features, as demonstrated by Equation (13). For the sake of limiting the computational burden, a maximum number of matched features has been set for the presented simulation tests. In particular, two cases have been run with $n_f = 20$ and $n_f = 100$, alternatively. The reader is suggested to refer to OpenCV documentation for the details of the implementation [25].

*5.3. Trajectory Estimation*

The navigation algorithm has been implemented in Matlab-Simulink software. A *python* interface is used to link the output of the image processing to the input of the

navigation module. The sample trajectory used for testing is the one described in Section 5.1. The proposed navigation system provides an estimate of the translation states only, as in [1], but relies on attitude determination to identify the camera pointing direction: then, attitude estimation errors could have an impact on navigation performances. The spacecraft rotational dynamics are not simulated: the navigation camera is assumed to maintain a nominal nadir pointing, while a Gaussian noise with standard deviation $\sigma = 1°$ is added on the three Euler angles to represent attitude determination errors. The altimeter measurements are synthetically generated by randomly perturbing the vertical groundtruth position with a Gaussian noise. A standard deviation of 1% of the current altitude is assumed, reflecting the actual behavior of the laser altimeter technology. The whole model is implemented in a *Matlab-Simulink* environment, with altitude measurements generated at frequency 8 Hz. The navigation filter parameters are listed in Table 1. The initial condition is given as a perturbed state with respect to the groundtruth with a Gaussian noise with variance $\sigma^2 = 10^4 \text{ m}^2$, which represents a potential initialization error when the navigation switches from absolute to relative mode. The estimation error remains bounded with a mean horizontal error and vertical error of less than 200 and 100 m, respectively, representing less than the 0.5% of the position vector magnitude. As showed in Figure 11, the number of matched features pairs that is processed by the algorithm has an impact only on the estimated covariance. In other words, this means that the number of features mainly influences the robustness of the filter without strongly affecting the accuracy of the estimation. This is partially expected, as each epipolar equation is a standalone geometric constraint. In principle, a minimum number of correspondences is needed for static (i.e., without navigation filters) pose estimation, as discussed in Sections 1.1 and 1.2; nevertheless, increasing the number of matches affects the covariance estimate at the cost of higher computational burden.

**Table 1.** Simulation parameters.

| Parameter | Value | Description |
|---|---|---|
| $n_{f,max}$=20, 100 | 50 | Maximum number of processed matched features |
| $\mathbf{P}_0$ | $\text{diag}(10^4\mathbf{I}_{3\times3}, 10^0\mathbf{I}_{3\times3})$ | Initial Covariance Matrix |
| $\mathbf{R}_{k,elem}$ | $10^0\mathbf{I}_{2\times2}$ | Elementary crater localization error covariance |
| $\mathbf{R}_{k,alt}$ | $10^2$ | Elementary altimeter error variance |

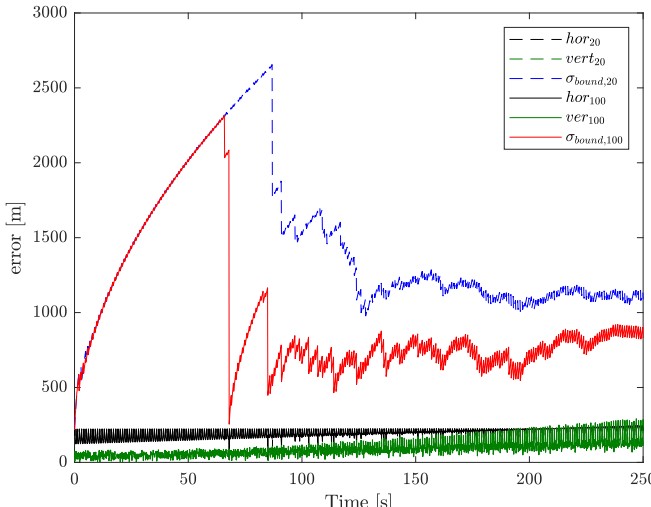

**Figure 11.** Numerical results for a sample landing trajectory with different maximum number of processed features.

*5.4. Monte Carlo Analysis*

To analyze the robustness of the algorithm, a Monte Carlo analysis is performed to explore the uncertainties that typically affect the landing scenario and the filter tuning. In

particular, the uncertainties distribution around the nominal values (cfr. Table 1) of the analyzed variables are reported in Table 2. The analyzed variables are deemed to be the most uncertain in a realistic deployment. They consist of the position and velocity initial conditions $\vec{x}_0 = [\vec{r}_0, \vec{v}_0]^\intercal$, initial covariance matrix $\mathbf{P}_0$, and the elementary measurement covariance matrix $\mathbf{R}_k$ for features localization and altimeter.

**Table 2.** Monte Carlo variables distribution assumptions around nominal values. For the matrices, each component is perturbed with the same random distribution.

| Parameter | $\sigma$ | Distribution |
|:---:|:---:|:---:|
| $\vec{r}_0$ | 100 m | Gaussian |
| $\vec{v}_0$ | 0.1 m/s | Gaussian |
| $\mathbf{P}_0$ | $\text{diag}(10^3 \mathbf{I}_{3\times3}, 10^{-1}\mathbf{I}_{3\times3})$ | Gaussian |
| $\mathbf{R}_{k,elem}$ | $10^{-1}\mathbf{I}_{2\times2}$ | Gaussian |
| $\mathbf{R}_{k,alt}$ | $10^1$ | Gaussian |

The aim of the Monte Carlo analysis is twofold:

1. *Objective 1*: to analyze global results in terms of the landing trajectory mean error to estimate product confidence levels.
2. *Objective 2*: to analyze the estimation error throughout the trajectory to evaluate filter robustness and convergence.

The number of runs of the statistical analysis is selected to represent the global response of the system to the assumed uncertainties, according to Hanson [26]. The goal of the analysis is to estimate the mean estimation error value through the landing trajectory. Therefore, in terms of statistical analysis, the goal is to estimate the mean estimation error bound within a box at ~99.73% of probability. It is important to note that, in the order statistics approach (*Objective 1*), the necessary number of samples does not depend on how many uncertain variables are varied. The standard error of the mean is used to assess the accuracy of the Monte Carlo analysis. The standard error of the mean is calculated using the sample standard deviation:

$$SE_{\bar{e}} \approx \frac{\sigma_e}{\sqrt{n}}. \tag{31}$$

The standard error of the sample mean is an estimate of how far the sample mean is likely to be from the population mean, whereas the standard deviation of the sample is the degree to which individuals within the sample differ from the sample mean. As reported in Table 3, the standard error of the mean is lower than 1 m using 100 samples, which is considered as acceptable. Table 3 reports the mean and the ~99.73% probability with $3\sigma$ bound for the filter performance.

**Table 3.** Monte Carlo results for landing trajectory mean estimation error.

| Metric | $\mu_e$ | $3\sigma_e$ [m] | $SE_{\bar{e}}$ [m] |
|:---:|:---:|:---:|:---:|
| horizontal error | 193.9 | 27.3 | 0.9 |
| vertical error | 97.8 | 23.6 | 0.8 |

For *Objective 2*, the Monte Carlo runs are visualized in Figures 12 and 13. The $3\sigma$ bound derived from the filter covariance estimation is compared with the $3\sigma$ bound at each time step of the Monte Carlo population. The filter covariance estimate is larger than the Monte Carlo variance, meaning that the filter is being conservative in its performance. Furthermore, the variables taken into account in the Monte Carlo analysis are a subset of the uncertainties that the filter is experiencing in the simulated scenario (e.g., variable altimeter measurements errors as a function of the altitude). The conservative behavior is favorable for the robustness of the filter. The vertical error shows an estimated covariance lower than the Monte Carlo bound for a limited portion of the trajectory. This may be due to the selected nominal case to which the filter covariance estimation refers to. Nevertheless, in Figure 13b,

one can see that the filter covariance estimation still bounds correctly the estimation error of the whole Monte Carlo samples except for few error spikes. Figures 12b and 13b show a zoom of the overall plot to highlight the behavior of the different Monte Carlo samples.

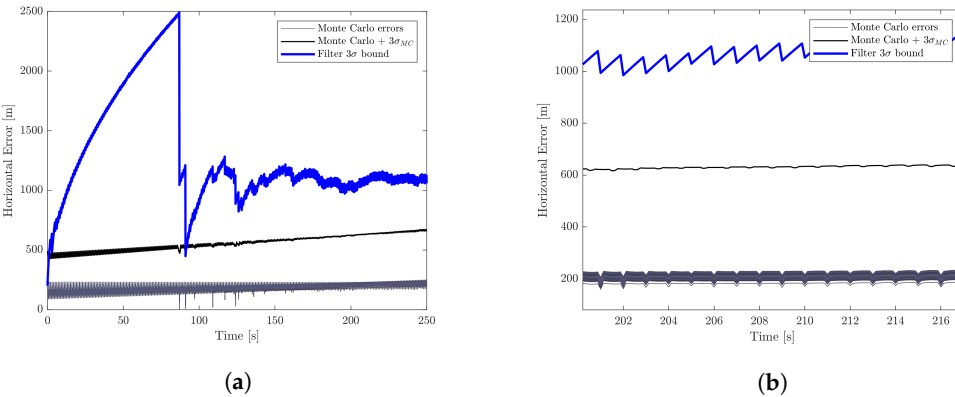

(**a**)                                                (**b**)

**Figure 12.** Monte Carlo runs for the landing trajectory. At each time step, the horizontal estimation error for each Monte Carlo sample is reported. The $3\sigma$ bound derived from the filter covariance estimation is compared with the $3\sigma$ bound at each time step of the Monte Carlo population. (**a**) Horizontal error—Monte Carlo runs. (**b**) Horizontal error—Monte Carlo runs: zoom.

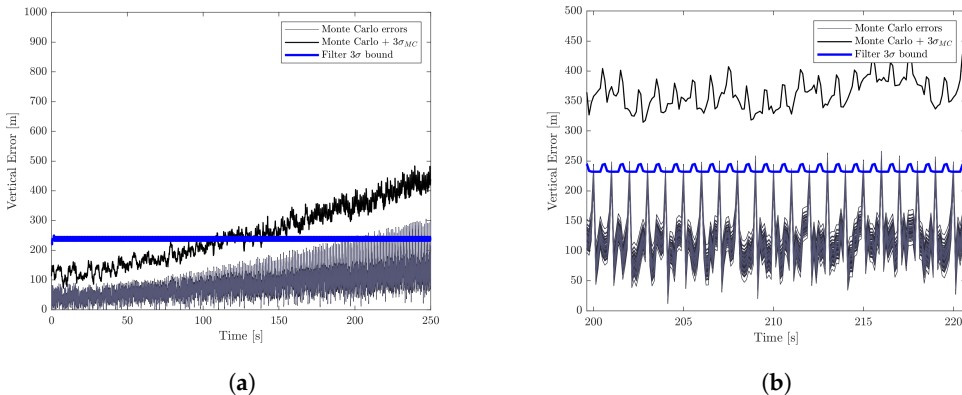

(**a**)                                                (**b**)

**Figure 13.** Monte Carlo runs for the landing trajectory. At each time step, the vertical estimation error for each Monte Carlo sample is reported. The $3\sigma$ bound derived from the filter covariance estimation is compared with the $3\sigma$ bound at each time step of the Monte Carlo population. (**a**) Vertical error—Monte Carlo runs. (**b**) Vertical error—Monte Carlo runs: zoom.

*5.5. Computational Time: Comparison with Essential Matrix Pose Recovery*

This work presents a filter formulation that uses the feature correspondences directly in the update step; as depicted in Figure 1, a valid alternative is to pre-process the matched features to retrieve the pose from perspective geometry relationships. In particular, as discussed in Section 1.2, the 5-point algorithm can be used to find the essential matrix from a set of features correspondences. Once the essential matrix is determined, it can be decomposed using SVD decomposition. In general, four possible poses exists for a given **E**. Thus, the algorithm requires to verify possible pose hypotheses by performing the chirality check. The chirality check consists of the verification that the triangulated 3D points have positive depth [27]. By decomposing E, one only obtains the direction of the translation, so the function returns the translation unit vector. The altimeter measurement is used to scale the pose estimation.

To be consistent in the comparison, taking as reference Figure 1, only the relevant steps need to be timed. They consist of:

- IEKF:
  - Implicit epipolar constraint (Section 4.2);

- Correction step (Section 4.3): this step would be executed in all the presented methods, but its execution time is influenced by the size of the Kalman gain matrix, measurement matrix, and measurement covariance matrix. Hence, it is reasonable to maintain its contribution in the required computational time for the IEKF where the sizes of those matrices are not negligible.

- 5-point + SVD decomposition:
  - 5-point algorithm solver [27], including RANSAC for outlier rejection;
  - SVD decomposition of the essential matrix;
  - Chirality check [27];
  - Measurement scaling.

The summary results for the computational times are reported in Table 4. The algorithms have been run in a Intel© Core™ i7-6500U CPU @ 2.50 GHz, thus the numbers in Table 4 are only for relative comparison and they are not representative of any on-board implementation. The histogram for the elapsed times for the 5-point + SVD decomposition method is reported in Figure 14. It can be noted that the distribution is quite spread with a significant standard deviation (cfr. Table 4) due to the preemptive RANSAC iterative algorithm. On average, the decrease in computational time using IEKF is roughly 75%.

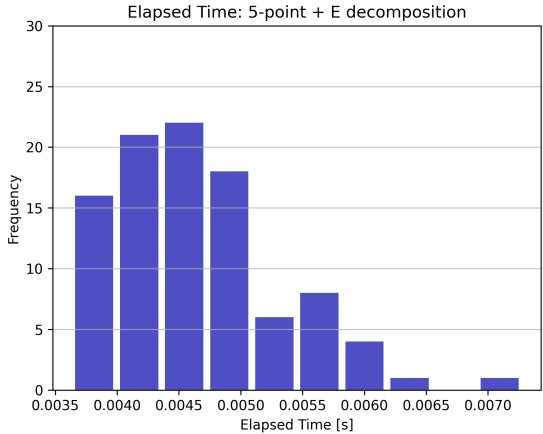

**Figure 14.** Distribution of execution times for the 5-point algorithm and pose recovery through SVD decomposition of the essential matrix and chirality check.

**Table 4.** Execution times for different approaches: IEKF (this work) and 5-point + SVD decomposition. The IEKF elapsed time refers to the computation of the implicit constraint and the relevant covariance matrices.

| Algorithm | Mean [s] | $\sigma$ [s] |
|---|---|---|
| IEKF-correction | 0.0012 | 0.0006 |
| 5-point + SVD | 0.0047 | 0.0015 |

## 6. Conclusions

This paper presented a method to perform relative (or local) terrain navigation using frame-to-frame features correspondences and altimeter measurements. In summary, the proposed image-based approach relies on the implementation of the implicit extended Kalman filter, which uses the epipolar constraint as the implicit measurement function. The navigation system relies also on a fast cycle with altimeter updates. This method allows to feed the navigation filter with the features coordinates directly to be fused with the altimeter measurements, without employing the pre-processing algorithms to retrieve the relative pose from features correspondences. For testing purposes, the altimeter acquisition was set to $f_{alt} = 8$ Hz, whereas the images acquisition $f_{opt} = 1$ Hz. Moreover, an extrapolation method has been developed to incorporate the intrinsic delay of the image processing

routine, which was set to 1 s. The algorithm has been tested on a sample landing trajectory delivering good results in terms of position estimation with respect to the pinpoint landing location. A systematic numerical testing campaign is foreseen to assess the robustness of the navigation approach.

**Author Contributions:** Conceptualization, S.S. and G.Z.; methodology, S.S.; software, S.S. and G.Z.; validation, S.S., M.P. and A.B.; formal analysis, S.S.; investigation, S.S.; resources, S.S.; data curation, S.S.; writing—original draft preparation, S.S.; writing—review and editing, S.S.; visualization, S.S.; supervision, P.L. and M.L.; project administration, M.L.; funding acquisition, M.L. All authors have read and agreed to the published version of the manuscript.

**Funding:** The presented work was funded by the European Space Agency, under the ESA Contract No. 4000129466/19/NL/CRS.

**Data Availability Statement:** Not applicable.

**Acknowledgments:** The authors would like to acknowledge the support from the AIVIONIC consortium: DEIMOS Engenharia S.A. (Portugal) as prime contractor; AIKO SRL (Italy), Politecnico di Milano (Italy), Fortiss Gmbh (Germany), Ubotica (Ireland), and DEIMOS Space S.L.U. (Spain).

**Conflicts of Interest:** The authors declare no conflict of interest.

## Abbreviations

The following abbreviations are used in this manuscript:

| | |
|---|---|
| FOV | Field Of View |
| GRS | Ground Reference system |
| IEKF | Implicit Extended Kalman Filter |
| LS | Landing Site |
| ME | Mean Earth/Polar Axis |
| PD | Power Descent |
| PM | Pitch Maneuver |
| PO | Parking Orbit |
| SVD | Singular Value Decomposition |

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
