# Peer review of "Implicit Extended Kalman Filter for Optical Terrain Relative Navigation Using Delayed Measurements"

_aerospace, doi:10.3390/aerospace9090503_

Round 1
Reviewer 1 Report
This paper proposes a novel method for relative terrain navigation using optical measurements and an altimeter in an implicit extended Kalman filter framework.
- Even though the authors mention that "novelty lies on the processing of the frame-to-frame matched features" irrespective of the dynamics. Wont the processing be affected if the dynamics employed were to be nonlinear? If the accurate dynamics are used wouldn't that affect the real-time performance of the nav stack?
- Just a very minor comment: the word "indeed" is over used.
- Figure 7: a,b,c, subplots are cut off.
- Equation 14: C_k-1,k = C(m_k-1,m_k-1) => C(m_k-1,m_k) ?
Reviewer 2 Report
This manuscript does not offer anything new in the scope of IEKF or application. Trajectory estimation using camera and constraints have been done many times in the literature. Furthermore, a single simulation (figure 10) is not enough to convince the readers the results are sound. A Monte Carlo simulation is the minimum to show how the algorithm works in various scenarios.
Reviewer 3 Report
In summary, the paper proposed a method to perform relative (or local) terrain navigation using frame-to-frame features correspondences and altimeter measurements, based on implementing the implicit extended Kalman filter. The modified extrapolation method is utilized to address the delay of the matching process. The research is practical and simulations are presented. There are some problems, which need to be addressed by the authors.
1. One of the critical contributions of this paper is to deal with the navigation problem using the IEKF, and the author should explain the advantages of the IEKF over other filters, for example, other Kalman Filters.
2. In the simulation part, some comparisons between the proposed method and the existing methods could be done, to show the superiority of the IEKF trajectory estimation method, in terms of accuracy, computational burden, and so on.
3. There are compile problems in fig 7.
4. Texts in Figs 1 and 2 are too small and vague.
